# Salivary biomarkers of tactical athlete readiness: A systematic review

Bryndan Lindsey[1☯], Yosef Shaul[2☯], Joel Martin[2,3]*

1 Research and Exploratory Development Department, Johns Hopkins Applied Physics Laboratory, Laurel, Maryland, United States of America, 2 Sports Medicine Assessment Research & Testing (SMART) Laboratory, George Mason University, Virginia, United States of America, 3 Center for the Advancement of Well-Being, George Mason University, Fairfax, Virginia, United States of America

☯ These authors contributed equally to this work.
* jmarti38@gmu.edu

## Abstract

Tactical athletes must maintain high levels of physical and cognitive readiness to handle the rigorous demands of their roles. They frequently encounter acute stressors like sleep deprivation, muscle fatigue, dehydration, and harsh environmental conditions, which can impair their readiness and increase the risk of mission failure. Given the challenging conditions these athletes face, there is a vital need for non-invasive, rapidly deployable point-of-care assessments to effectively measure the impact of these stressors on their operational readiness. Salivary biomarkers are promising in this regard, as they reflect physiological changes due to stress. This systematic review aims to investigate salivary markers as potential indicators for readiness, specifically focusing on their sensitivity to acute stressors like sleep deprivation, dehydration, environmental factors, and muscle fatigue. A search was conducted using the Preferred Reporting Items for Systematic Review and Meta-Analyses (PRISMA) guidelines (PROSPERO; registration #: CRD42022370388). The primary inclusion criteria were the use of a quantitative analysis to assess salivary biomarkers changes in response to acute stressors. Risk of bias and methodological quality were evaluated with the modified Downs and Black checklist. Hormonal salivary biomarkers were the most commonly studied biomarkers. Muscle damage and fatigue were the most frequently studied acute stressors, followed by sleep deprivation, multiple stressors, dehydration, and environmental. Biomarkers such as creatine kinase, aspartate aminotransferase, uric acid, cortisol, testosterone, and the testosterone to cortisol ratio were indicative of muscle damage. Dehydration influenced osmolality, total protein, flow rate, and chloride ion concentrations. Sleep deprivation affected proteins, peptides, and alpha-amylase levels. Environmental stressors, such as hypoxia and cold temperatures, altered cortisol, pH, dehydroepiandrosterone-sulfate (DHEA-s), and salivary IgA levels. The current body of research highlights that various salivary biomarkers react to acute stressors, and proteomic panels appear promising for predicting physical and cognitive outcomes relevant to the operational readiness of tactical athletes.

**Data availability statement:** All relevant data are within the paper and its Supporting Information files.

**Funding:** The author(s) received no specific funding for this work.

**Competing interests:** The authors have declared that no competing interests exist.

## Introduction

The occupational physiological demands of tactical athletes (TA), such as soldiers and emergency responders (police officers, fire fighters, emergency medical technicians, etc.) during stressful training and operational environments are comparable to that of competitive sporting athletes [1] and are often extreme, including extended periods of high energy expenditure, dehydration, and limited sleep [2]. The subsequent strain on physiology can cause fatigue [3] and degrade health and performance [4]. For the military and other tactical organizations, insufficient recovery and excessive training loads can increase the likelihood of injury [5] and may reduce operational readiness [5].

From a human performance perspective, operational readiness refers to the state of condition of individuals, which enables them to carry out their occupational duties effectively and efficiently [6]. Physical, cognitive, and emotional states can be altered rapidly due to the aforementioned stressors potentially impairing one's level of operational readiness [6–8]. Traditional evaluation of TA operational readiness includes medical examinations, fitness assessments, and subjective self-assessment questionnaires, which can be imprecise, time-consuming, and often fail to predict performance outcomes [8]. Therefore, strategies to measure physiological readiness quickly and effectively in the field after exposure to occupational stressors are desired to objectively monitor TA status in-situ [9].

Stressors that impact operational readiness were outlined by a panel of military experts in 2013 and include physical fatigue, extreme environmental conditions (e.g., cold/hot temperatures, high altitude), altered psychological status (e.g., stress, anxiety, depression), and sleep quantity and quality [9]. When stressor events occur, sympathetic stimulation of the autonomic nervous system releases adrenaline and noradrenaline [10]. A secondary, slower response also occurs from stimulation of the hypothalamic-pituitary-adrenal (HPA) axis which releases glucocorticoids such as cortisol [11]. Therefore, it has been hypothesized that measurement of such signaling molecules that induce physiological changes associated with acute stress may have diagnostic utility for TA operational readiness assessment [12]. Training induced changes in circulating levels of cortisol and testosterone show potential as markers of excessive training stress, physiological strain, and inadequate recovery in non-military populations [1,13]. These hormones have also been found to be sensitive to various forms of military training stress, with cortisol concentrations increasing during and following training [14,15] and total and free testosterone concentrations decreasing [16]. Additionally, other biomarkers have been associated with these stressors such as dehydroepiandrosterone (DHEA) with physical stress [17], salivary pH with hypoxia from altitude [18], alpha amylase with sleep deprivation [19], and osmolality with dehydration [20], amongst others. Therefore, measurement of the biochemical signatures of stress shows promise to objectively quantify TA readiness and inform decision making in operational settings [7].

Biomarkers can be analyzed from various media including blood, urine, and/or saliva [7]. Serum-based biomarkers like cortisol [21], dehydroepiandrosterone (DHEA) [22], epinephrine and norepinephrine [23], neuropeptide-Y (NPY) [24],

and brain derived neurotropic factor (BDNF) [25] have shown to be both responsive and potentially protective to military related stressors like military training [21,23,25], while inflammatory cytokines like interleukin 6 (IL-6) [26], tumor necrosis factor (TNF-α) [26], and interleukin 10, (IL-10) [25] are well known to be responsive of exercise and training stress. However, although blood has most often been used as the source of measurable biomarkers in diagnostic studies, there are numerous advantages to the use of saliva, most notably including its ease of collection (e.g., non-invasive) and lower risk of transmission [27]. In addition, saliva contains many analytes affected by a range of physiological and pathological stressors [28,29] which have been highly correlated with those in blood [30,31], have shown good reproducibility and changes across time [32], with research showing that analytes like salivary cortisol are associated with reduced performing after military training [33]. In fact, research has demonstrated strong positive correlations with serum levels for many of the more commonly studied salivary biomarkers like cortisol (r = 0.92) [34], testosterone (r = 0.65) [35], and DHEA (r = 0.86) [36]. Despite this, the use of saliva-based biomarkers as a diagnostic signal has yet to become widely adopted, likely because although most analytes detected in blood are also found in saliva, their levels are diminished [37]. However, due to the small quantities of fluid needed for extraction of biomarkers from saliva, the utility of the media still warrants investigation.

At present it is necessary to characterize the body of literature which has reported on the change in salivary biomarkers after a variety of acute stressors to guide future research. Therefore, this systematic review seeks to explore and identify salivary markers as potential candidates for diagnosing state of readiness. The main aim was to identify salivary markers sensitive to acute stressors of sleep deprivation, dehydration, environmental, and muscular fatigue. Secondary aims were to: (i) to identify salivary markers that are sensitive to physical readiness and (ii) to identify salivary markers that are sensitive to cognitive readiness. The systematic review aims hold practical importance as they provide valuable insights into the relationship between salivary biomarkers and domains of operational readiness following exposure to common acute stressors. By characterizing the existing literature and identifying salivary markers sensitive to stressors and cognitive readiness, this review can guide future research and contribute to the development of non-invasive diagnostic tools for monitoring operational readiness, potentially leading to reduced risks in high-demand environments.

## Materials and methods

### Review registration

The study search criteria, screening, selection, data extraction, and quality assessment were defined a priori and informed by the Preferred Reporting Items for Systematic Reviews and Meta-Analyses guidelines (PRISMA) [38]. The systematic review was registered in the International Prospective Register of Systematic Reviews (PROSPERO; registration #: CRDXXXXXXXXXXX – BLINDED).

### Eligibility criteria

The following inclusion criteria were applied to the review: a) published within the past 30 years, b) English language only, c) peer reviewed articles, d) full text availability, e) human subjects research, f) healthy adults aged 18–60 years, g) investigated salivary biomarkers as determinants of outcomes of 'readiness' in domains that include athletic performance, aerobic and strength performance, or performance of cognitive tasks, h) incorporated acute stressors that included sleep deprivation, dehydration, environmental, or muscle fatiguing activities, and i) quantified a change in biomarker due to an acute stressor. Studies investigating biomarkers of disease (Alzheimer's, diabetes, Parkinson's, dementia, cancer) were excluded.

### Information sources

In line with recommendations for comprehensive literature coverage and recall [38], four databases—PubMed, CINAHL, SportDiscus, and Google Scholar —were searched for relevant publications [39]. Google Scholar was searched using

the Publish or Perish software [40]. The searches were conducted between January and February 2024. Additionally, the reference lists of the retrieved articles were reviewed to identify any further potentially relevant studies not captured by our initial search strategy [41].

## Search strategy and study selection

A preliminary examination of topic-related studies was conducted to identify key search terms and categories. A set of keywords and phrases using Boolean logic were then used to search relevant databases and identify original research for review. Where possible, filters such as years since publication and language, were applied to meet eligibility criteria. If a filter did not exist, eligibility was determined manually through screening of title and abstract. The list of search terms and filters used to search the selected databases are detailed in S1 Table.

Due to search string length limitation, the following string was applied for Google Scholar: saliva AND marker AND (athletic | fitness | strength | aerobic | cognitive | psych* | exercise) AND performance AND (sleep | injury | dehydration | "acute stress*") -elderly -Alzheimer -Parkinson -dementia -Insomnia -cancer -diabetes -patient -child. Filters included English language and past 30 years. Following previously published recommendations, the top 500 citations were exported instead of top 200 [42].

Two authors (YS and BL) conducted separate reviews on the four electronic databases. Full search results were exported to SciWheel reference and citations manager (SAGE Publications, London, UK). Duplicates were searched for and removed on SciWheel. Initial screenings of the titles and abstracts were independently conducted by two authors (YS, BL) to assess whether eligibility criteria were met. In cases of a disagreement, a third author (JM) was consulted. The full list of articles retrieved is provided in S2 Table.

## Data extraction and synthesis

For all eligible studies, data were extracted and tabulated independently by two authors (YS, BL). Data was extracted from all eligible studies into a spreadsheet based on the Cochrane Consumers and Communication Review Group's data extraction template [43]. Extracted information included i) author, ii) year, iii) country, iv) title, v) participants, vi) study design, vii) study description, viii) salivary biomarkers and collection method, ix) salivary biomarker quantification method, x) acute stressors, xi) paired readiness outcome measures, xii) quantified change in biomarker concentrations, and xiii) key findings. Statistically significant changes in readiness outcomes due to acute stressors were noted. Study design was classified based on guidance provided by the Centre for Evidence-Based Medicine [44] with the addition of the definition of 'quasi-experimental' for non-randomized intervention studies. Following the data extraction, a qualitative synthesis was performed regarding salivary biomarkers by acute stressor. Per the aims of the review, all studies meeting the eligibility criteria were synthesized qualitatively with counts of significant changes in biomarkers per stressor tabulated and reported in text.

## Risk of bias and quality assessment

The included studies were evaluated for risk of bias and quality using the Downs and Black 27-item checklist, which assesses methodological quality for both randomized and non-randomized studies. The checklist includes ratings for the following constructs: reporting, external validity, bias, confounding, and power [45]. Of the 27 items, 25 are scored on a scale of 0–1 point, with "yes" indicating the criterion is met (1 point) and "no/unable to determine" indicating it is not (0 points). Item 5, which assesses the detailing of confounders, scores as follows: "yes" equates to 2 points, partially to 1 point, and no to 0 points. Item 27 assesses statistical power on a scale up to 5 points. For this review, it was modified to score "yes" (1 point) if the study detailed sufficient power to detect a clinically important effect, and "no/unable to determine" (0 points) otherwise [45]. This modification has been used in previous reviews [46] to reduce potential bias from ambiguous wording [47].

A total score was computed as the sum of the 27 items on the Downs and Black Checklist, and a percentage was calculated by dividing the total score by the total possible points. These percentage scores were then used to determine the overall quality of the studies. In this review, Kennelly's [48] proposed grading system was modified from the original Downs and Black checklist, changing the total possible points from 32 to 28. Consistent with previous reviews [46], studies scoring below 46.9% were classified as having 'poor' methodological quality, those scoring 46.9–62.5% as having 'fair' methodological quality, and those scoring above 62.5% as having 'good' methodological quality.

## Results

### Search results

The PRISMA search diagram (Fig 1) depicts the results of the literature search, screening, and selection process. In total, there were 2,386 search results, of which 1,207 duplicates were removed, yielding 1,179 studies. Title and abstract screening yielded 95 studies that were screened for full text. Finally, 47 of these met the eligibility criteria while 2 more were added during a backward/forward search to yield the final 49 included studies.

### Methodological quality

Article scoring on the Downs and Black checklist is displayed in Table 1. Included articles ranged from 'fair' (n=22) to 'good' (n=27) quality ratings with an overall mean(±SD) score of 18.5±1.3. On average, articles scored well on reporting (80%) and internal validity - bias (73%), but poorly on external validity (33%) and internal validity - confounding (57%). Only 5 of 49 studies reported performing a power analysis to ensure adequate sample size to detect significant differences. Specifically, most studies scored poorly (less than 25% scoring 'yes') on items 8, 14, 15, 23, 24, and 27. Additionally, studies also only performed fair (less than 70% scoring 'yes') on items 5, 21, and 22.

### Characteristics of included studies

**General Characteristics.** Details and key findings extracted from the included studies are reported in Fig 2 and S3 Table. A majority (67%) of the research was conducted in the USA, Australia, and the United Kingdom. Most (71%) were published within the last 10 years and none were older than 20 years. The average sample size of the included studies was 23 out of a total 1,118 participants across all studies with ages in included samples ranging from 18–52 years. A diverse range of participants were represented in the included articles (Fig 2A), with 9 studies involving military participants [17,33,49–55], as well as a variety testing athletes of sports like rugby [56–60], soccer [61–65], running [66–68], rock climbers [69,70], weight-lifting [71,72], and cycling [20,73–75] amongst other sports [76,77]. Many also simply tested healthy adults [1,18,19,78–85]. In addition, 2 studies considered emergency physicians and investigated the exposure of health care personnel to acute stressors and possible implications on their performance and decision making [86,87]. However, of the 49 studies, only 13 included both sexes, with a total of 194 males and 101 females in these studies.

**Experimental methodology.** Details on the designs, measured biomarkers, performance measures, and key findings are reported in S3 Table. Twenty-two out of the 49 studies were prospective cohort studies in that they attempted to describe how salivary markers change in response to stressors that are seen in a naturalistic setting but lacked comparative control groups [17,33,49–51,53,54,57–60,64–66,68,70,77,86–90]. Twelve studies were quasi-experimental in that the researchers measured response to imposed stressors pre-post [19,67,69,71,74,76,78,79,82,83,85,91]. Eleven studies were randomized cross-over designs [18,20,56,61,63,72,73,80,81,84,75] while 3 randomized participants into one of multiple stressor groups (parallel design) [52,55,62]. Only 1 study randomized participants into a stressor and true control group [92]. Three of the studies reported a predictive model of a participant's outcome based on levels of salivary biomarkers [19,58,73]. Of these 3, a model was constructed to predict performance following fatigue [73] was

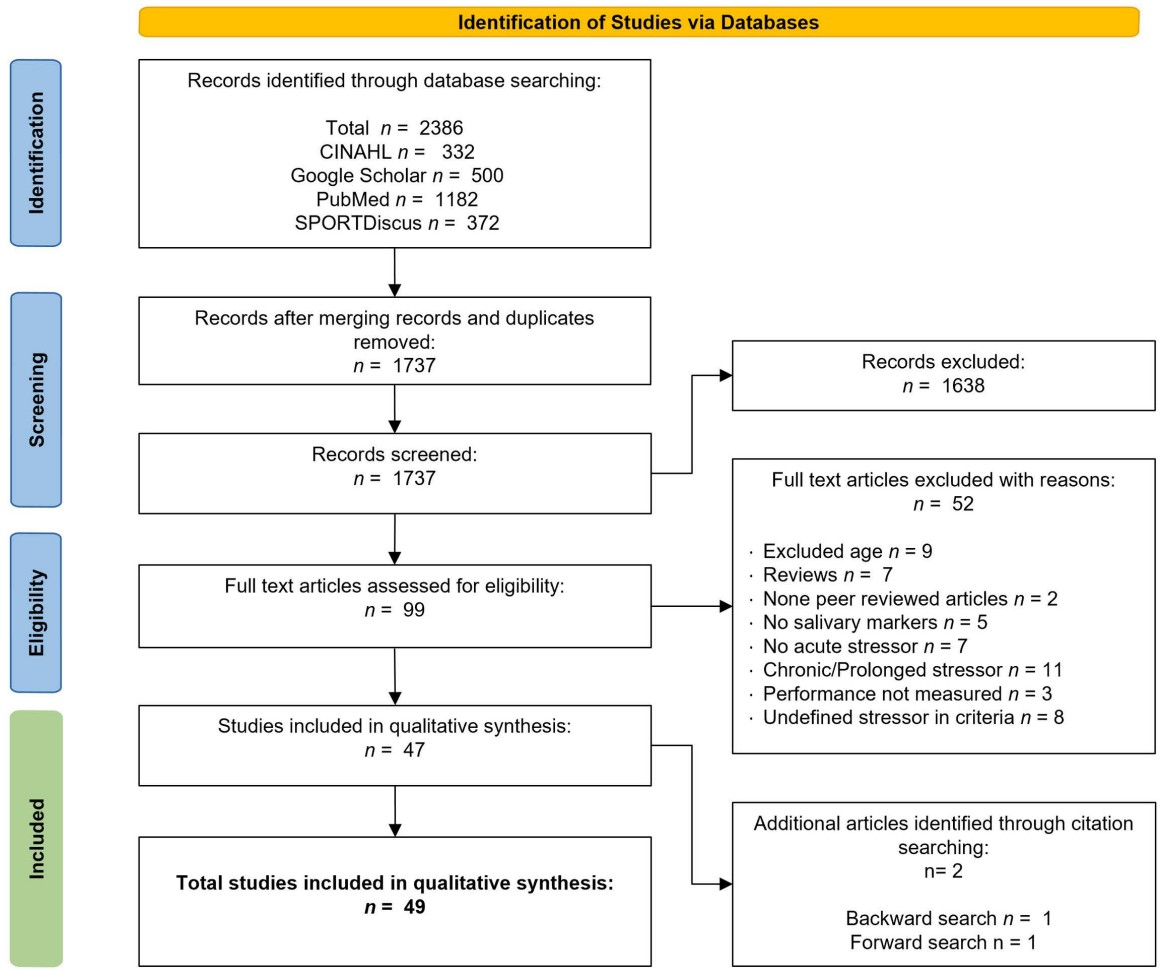

**Fig 1. Search flow diagram for the systematic review.**

later validated on a different population with a fatigue-inducing stressor (sleep deprivation) [92]. Distribution of key methodological characteristics of identified studies are visualized in Fig 2B-D.

Methodological details and reported outcomes for all identified studies are summarized in S3 Table. Fatigue, followed by sleep deprivation, multi-stressors, dehydration, hypoxia via altitude, and environment (cold stress) were the most studied acute stressors in the studies included (Fig 2C). As stated previously, stressors were determined *a priori* to be classified into 5 categories: sleep deprivation, muscle damage (fatigue), dehydration, environmental stress, and combined stressors (studies that involved more than one of the prior listed stressors). Additionally, there was a tendency to include specific populations relevant to the stressor. For example, altitude induced hypoxia studies involved trekkers [90] or aviators [55], two populations typically exposed to high altitude. Interestingly, a few studies have included sport game matches as stressors and attempted to predict match outcome (wins/losses) [58,59,88]. Of the 12 studies investigating biomarkers of sleep quality, 4 induced full sleep deprivation [78,92,61,19] while the rest were partial [49,57,68,87,86,93,94]. Of the 26 studies measuring biomarker response to muscle damage and/or fatigue 8 used exercise protocols [20,33,71–73,79–81], 4 involved military training [50–53], while the remaining 14 investigated biomarker response to sports training [58–60,62–66,69,70,74,77,88,89]. Of the 5 studies

**Table 1. Study risk of bias and quality assessment.**

| Author | Reporting | | External Validity | | Internal Validity (Bias) | | Internal Validity (Confounding) | | Power | | Overall | |
|---|---|---|---|---|---|---|---|---|---|---|---|---|
| | Item (n=10) Sum | % | Item (n=3) sum | % | Item (n=7) sum | % | Item (n=6) sum | % | Item 27 | Total | % | Rating |
| Cook et al. (2011) | 9 | 82 | 1 | 33 | 6 | 86 | 5 | 83 | 0 | 21 | 75 | Good |
| Chennaoui et al. (2009) | 9 | 82 | 1 | 33 | 5 | 71 | 4 | 67 | 0 | 19 | 68 | Good |
| Donald et al. (2017) | 8 | 73 | 1 | 33 | 7 | 100 | 3 | 50 | 1 | 20 | 71 | Good |
| Machi et al. (2012) | 9 | 82 | 1 | 33 | 5 | 71 | 4 | 67 | 0 | 19 | 68 | Good |
| Mantua et al. (2020) | 8 | 73 | 1 | 33 | 5 | 71 | 4 | 67 | 0 | 18 | 64 | Fair |
| Michael et al. (2013) | 9 | 82 | 1 | 33 | 5 | 71 | 3 | 50 | 0 | 18 | 64 | Fair |
| Pajcin et al. (2017) | 9 | 82 | 1 | 33 | 6 | 86 | 2 | 33 | 0 | 18 | 64 | Fair |
| Serpell et al. (2019) | 9 | 82 | 1 | 33 | 5 | 71 | 4 | 67 | 0 | 19 | 68 | Good |
| Summers et al. (2021) | 9 | 82 | 1 | 33 | 5 | 71 | 3 | 50 | 0 | 18 | 64 | Fair |
| Taylor et al. (2017) | 10 | 91 | 1 | 33 | 5 | 71 | 4 | 67 | 0 | 20 | 71 | Good |
| Xu et al. (2018) | 10 | 91 | 1 | 33 | 5 | 71 | 4 | 67 | 0 | 20 | 71 | Good |
| Bellar et al. (2017) | 9 | 82 | 1 | 33 | 5 | 71 | 3 | 50 | 0 | 18 | 64 | Fair |
| Bonato et al. (2020) | 10 | 91 | 1 | 33 | 5 | 71 | 4 | 67 | 0 | 20 | 71 | Good |
| Chen et al. (2017) | 9 | 82 | 1 | 33 | 5 | 71 | 2 | 33 | 0 | 17 | 61 | Fair |
| Crewther et al. (2013) | 8 | 73 | 1 | 33 | 5 | 71 | 4 | 67 | 0 | 18 | 64 | Fair |
| Crewther et al. (2016) | 10 | 91 | 1 | 33 | 5 | 71 | 3 | 50 | 0 | 19 | 68 | Good |
| Crewther et al. (2018) | 9 | 82 | 1 | 33 | 5 | 71 | 4 | 67 | 0 | 19 | 68 | Good |
| Erskine et al. (2007) | 10 | 91 | 1 | 33 | 5 | 71 | 3 | 50 | 0 | 19 | 68 | Good |
| Gaviglio and Cook (2014) | 9 | 82 | 1 | 33 | 5 | 71 | 4 | 67 | 0 | 19 | 68 | Good |
| Gomez-Merino et al. (2003) | 8 | 73 | 1 | 33 | 5 | 71 | 4 | 67 | 0 | 18 | 64 | Fair |
| González-Hernández et al. (2020) | 9 | 82 | 1 | 33 | 5 | 71 | 3 | 50 | 0 | 18 | 64 | Fair |
| Magiera et al. (2018) | 8 | 73 | 1 | 33 | 5 | 71 | 3 | 50 | 0 | 17 | 61 | Fair |
| Magiera et al. (2019) | 8 | 73 | 1 | 33 | 5 | 71 | 3 | 50 | 0 | 17 | 61 | Fair |
| McLean et al. (2010) | 9 | 82 | 1 | 33 | 5 | 71 | 4 | 67 | 0 | 19 | 68 | Good |
| Merrigan et al. (2021) | 9 | 82 | 1 | 33 | 5 | 71 | 2 | 33 | 1 | 18 | 64 | Fair |
| Michael et al. (2012) | 7 | 64 | 1 | 33 | 7 | 100 | 4 | 67 | 0 | 19 | 68 | Good |
| Rutherfurd-Markwick et al. (2017) | 9 | 82 | 1 | 33 | 5 | 71 | 4 | 67 | 0 | 19 | 68 | Good |
| Slivka et al. (2010) | 9 | 82 | 1 | 33 | 5 | 71 | 4 | 67 | 0 | 19 | 68 | Good |
| Sparkes et al. (2020) | 9 | 82 | 1 | 33 | 5 | 71 | 4 | 67 | 0 | 19 | 68 | Good |
| Springham et al. (2021) | 10 | 91 | 1 | 33 | 5 | 71 | 4 | 67 | 0 | 20 | 71 | Good |
| Viana-Gomes et al. (2018) | 6 | 55 | 1 | 33 | 5 | 71 | 3 | 50 | 0 | 15 | 54 | Fair |
| McKetney et al. (2022) | 9 | 82 | 1 | 33 | 5 | 71 | 4 | 67 | 0 | 19 | 68 | Good |
| Akazawa et al. (2019) | 9 | 82 | 1 | 33 | 5 | 71 | 4 | 67 | 0 | 19 | 68 | Good |
| Crewther et al. (2020) | 8 | 73 | 1 | 33 | 5 | 71 | 4 | 67 | 0 | 18 | 64 | Fair |
| Fogt et al. (2009) | 10 | 91 | 1 | 33 | 5 | 71 | 5 | 83 | 1 | 22 | 79 | Good |
| Heydari et al. (2022) | 8 | 73 | 1 | 33 | 5 | 71 | 4 | 67 | 0 | 18 | 64 | Fair |
| Lieberman et al. (2005) | 8 | 73 | 1 | 33 | 5 | 71 | 3 | 50 | 0 | 17 | 61 | Fair |
| Tait et al. (2022) | 9 | 82 | 1 | 33 | 5 | 71 | 4 | 67 | 1 | 20 | 71 | Good |
| Taylor et al. (2007) | 7 | 64 | 1 | 33 | 5 | 71 | 3 | 50 | 0 | 16 | 57 | Fair |

*(Continued)*

**Table 1.** (Continued)

| Author | Reporting Item (n=10) Sum | % | External Validity Item (n=3) sum | % | Internal Validity (Bias) Item (n=7) sum | % | Internal Validity (Confounding) Item (n=6) sum | % | Power Item 27 | Total | Overall % | Rating |
|---|---|---|---|---|---|---|---|---|---|---|---|---|
| Julià-Sánchez (2013) | 9 | 82 | 1 | 33 | 6 | 86 | 3 | 50 | 0 | 19 | 68 | Good |
| Pontremolesi et al. (2012) | 10 | 91 | 1 | 33 | 5 | 71 | 2 | 33 | 1 | 19 | 68 | Good |
| Tsunekawa et al. (2022) | 9 | 82 | 1 | 33 | 5 | 71 | 4 | 67 | 0 | 19 | 68 | Good |
| Woods et al. (2011) | 9 | 82 | 1 | 33 | 5 | 71 | 4 | 67 | 0 | 19 | 68 | Good |
| Meléndez-Gallardo et al. (2022) | 9 | 82 | 1 | 33 | 5 | 71 | 2 | 33 | 0 | 17 | 61 | Fair |
| Muñoz et al. (2014) | 9 | 82 | 1 | 33 | 5 | 71 | 2 | 33 | 0 | 17 | 61 | Fair |
| Muñoz et al. (2017) | 10 | 91 | 1 | 33 | 5 | 71 | 2 | 33 | 0 | 18 | 64 | Fair |
| Walsh et al. (2004) | 8 | 73 | 1 | 33 | 5 | 71 | 3 | 50 | 0 | 17 | 61 | Fair |
| Walsh et al. (2004) | 8 | 73 | 1 | 33 | 5 | 71 | 3 | 50 | 0 | 17 | 61 | Fair |
| Walsh et al. (2018) | 9 | 82 | 1 | 33 | 5 | 71 | 4 | 67 | 0 | 19 | 68 | Good |

investigating biomarker response to environmental factors, 4 investigated the effects of varying altitudes during training [18,55,67,90], while 1 investigated the effects of training in the cold compared to normal temperature [75]. Of the 5 studies measuring response to dehydration, 3 measured biomarker response to exercise-induced (active) dehydration only [85,91,75], while 2 measured biomarker response to both active and passive dehydration [82,83]. Eight studies measured biomarker response to combined stressors, 6 of which examined military populations during training [33,52,53,94,51] while 2 involved exercise paired with another stress source like sleep efficiency [76], and sleep and contextual variables (e.g., crowd size, play time, number of matches) [89].

Measured and reported outcomes types are grouped and depicted in Fig 2. Panel D. Seven studies included sports or exercise skills as an outcome. These included hand measures like grip strength [70,80], countermovement jumps [60,63,80], isometric dynamometry [81] and rugby game skills [89,93]. Five studies included game matches, competition outcomes, or sports event completion or success, for example rugby match win or loss, as outcome measures [66,77,58,59,88]. Heart rate was another common performance outcome being measured in 4 studies [18,70,74,20], while 1 study also reported heart rate variability (HRV) [70]. Six studies included self-reported fatigue in their outcomes [17,60,64,70,73,92]. Three studies reported self-perceived muscle soreness [60,64,72] and 1 study has reported self-reported recovery [53].

Four studies considered pulmonary variables as performance measures, including pulmonary function test [66], ventilatory efficiency and oxygen consumption [79], and oxygen saturation [90,18]. Seven studies measured body mass change as an indicator of hydration during dehydration protocols [33,52,85,91,82,83,95]. Eight studies included serum-based biomarkers as measures of stress alongside salivary markers [50,65,69,71,18,82,83,95]. Two of these measured blood lactate levels [69,18], 2 measured muscles damage enzymes [71,65], a further 2 reported blood hormonal markers [50,95], while the final 2 reported serum osmolality [83,82]. Six studies reported sleep quality [57,60,62,64,76,94] and 3 articles reported urine measures (osmolality etc.) [82,83,95] as stress measures. Additionally, 1 study reported muscle thickness [72] while 1 study reported electroencephalogram measures [86].

In addition to physical stress measures listed above, 6 studies measured cognitive tests outcomes [33,76,78,87,92,19] while another 6 studies measured mood states [33,60,63,64,74,92]. Stress levels were measured in 4 studies [53,64,77,94]. Other measures like success on a driving simulation [19] and fatigue established by level of electroencephalogram (EEG) theta waves [86] were recorded in 4 studies. Seven studies did not include a paired performance

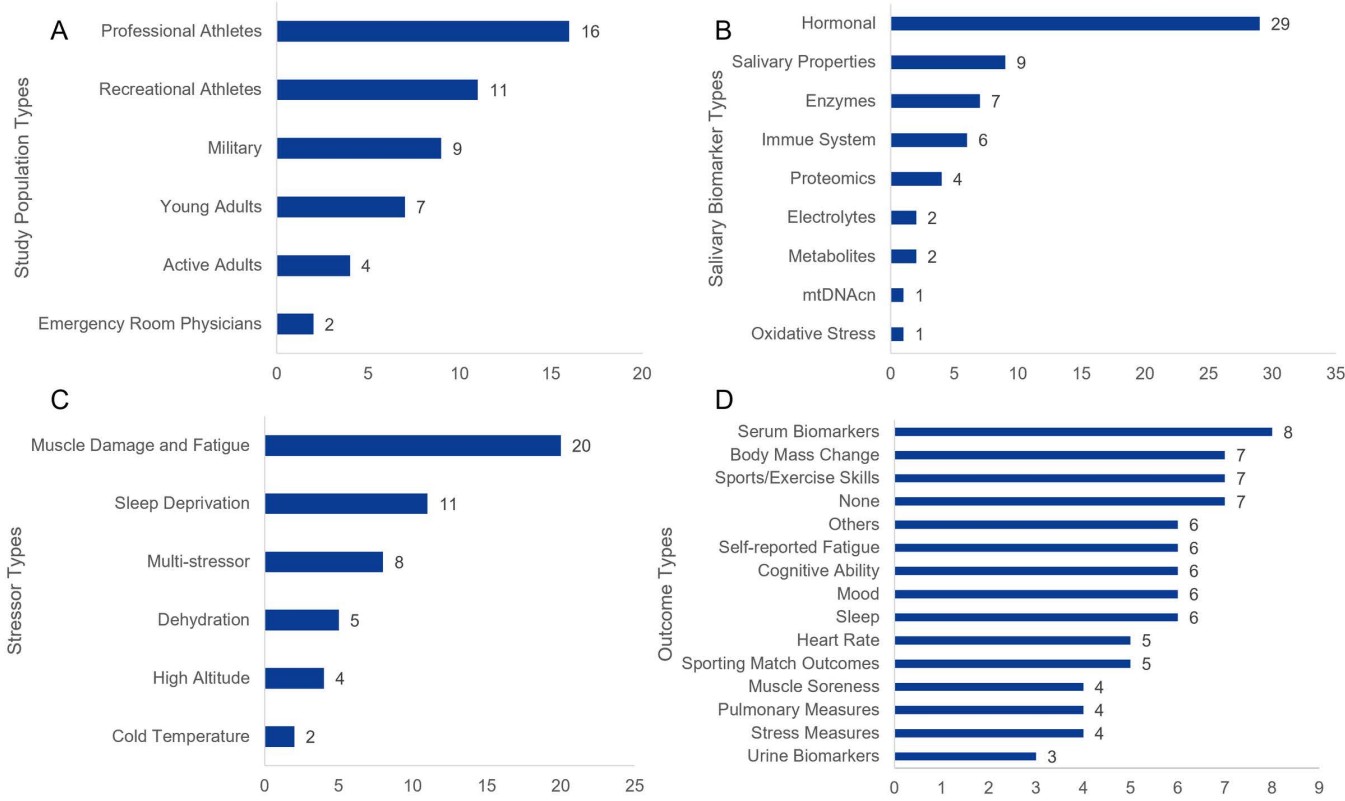

**Fig 2. Distribution of articles by key characteristics.** Panel A: Distribution of articles by population type studied. Panel B: Distribution of articles according to salivary biomarkers classification. Note: In 7 articles more than one type was examined. Panel C: Distribution of articles by acute stressors. Note: Muscle fatigue induced stressors include any exercise, endurance events, high intensity intervals, weightlifting, and rock climbing. The 'Multi-stressor' category includes combinations of different stressors, i.e., sleep deprivation and exercise. Panel D: Distribution of operational readiness outcomes measured. Note: In some studies, more than one outcome type was reported. 'Others' category includes outcomes like RPE, self-reported recovery, HRV, EEG, muscle thickness, driving simulation, proteomics, and metabolomics. Eight articles included self-reported measures (i.e., mood) with 3 of these reporting only self-reported measures. Abbreviations: mtDNAcn: Mitochondrial DNA Count Number; RPE: Rate of Perceived Exertion; HRV: Heart Rate Variability; EEG: Electroencephalogram.

outcome but instead only reported the change in level of salivary biomarkers in response to respective stressors [49,50,55,57,67,75,51].

Of the 49 studies, 33 used unstimulated methods to extract saliva while 6 used stimulated methods [33,54,67,76,19,86] with the remaining not reporting. Hormonal salivary biomarkers were the most commonly studied (Fig 2 Panel B), with 27 and 19 articles measuring cortisol and testosterone, respectively. The salivary properties of osmolality and flow rate were most commonly measured to study hydration [52,66,85,91,75,20,83,95]. The majority of studies (n=34) attempted to track two analytes at most (of the 34, 15 tracked a single analyte) while the remainder (n =15) tracked multiple analytes in response to stress. Excluding multi-omic studies, the greatest number of analytes studied was by Rutherfurd-Markwick et al. who measured 8 different analytes [20]. Several studies involved a clustering technique [51,73,86] that attempted to identify groups of proteins, metabolites, and peptides out of thousands identified in saliva, and relate the change in the expression of these groups to stress.

The identified studies employed a variety of analytical methods to quantify biomarkers, that can be broadly categorized into ten assay groups. The most frequently used were ELISA-based assays [50,53,55,60,64,66,68–70,72,87,89,75,61,20], being widely utilized in 15 out of 49 studies for their sensitivity and specificity in detecting salivary and plasma biomarkers

like cortisol and testosterone, while radioimmunoassay's were applied for hormone measurements like DHEAS and cortisol in 3 [17,33,79]. Enzyme immunoassays (n = 9) provided versatile applications for hormone and protein measurements [49,56,57,59,63,78,81,88,90]. Techniques involving chromatography and mass spectrometry were used in 5 studies [73,76,86,91,92] as they offer high precision in proteomic and metabolomic analyses. Similarly, general immunoassays (n = 5) [17,58,67,74,80] and osmometry-based assays (n = 5) [52,82–85] were essential for hormone quantification and osmolality measurements, respectively. Colorimetric and kinetic assays were used in 4 studies [19,62,65,77] to analyze enzyme activities, such as salivary alpha-amylase. Additionally, specialized techniques like spectrophotometry [71] and multi-omic approaches [51] highlighted innovative applications in biomarker detection.

The reported sensitivities and coefficients of variation (CVs) from the studies highlight the precision and reliability of the assays utilized. ELISA-based assays demonstrated high sensitivity, with cortisol detection limits as low as 0.007 µg/dL [61] and testosterone assays capable of detecting as little as 3.67 pmol/L [87]. Many assays reported intra-assay CVs below 10% and inter-assay CVs below 15%, indicating consistent performance across repeated measures. Enzyme immunoassays showed similar high sensitivity and low variation, further establishing their utility in precise biomarker quantification.

## Biomarker Responses to Acute Stressors

**Muscle Damage and Physiological Fatigue.** Twenty-six studies investigated salivary biomarker response to stressors that can potentially cause muscle damage and fatigue such as military training, weight-lifting, rock climbing, ultra-endurance events, and high intensity interval training [33,50,52,53,60,62–64,66,69–72,74,77,79–81,89,20,51,58,59,88,65,73]. The biomarkers studied in response to these stressors included cortisol [53,60,62–64,69,70,72,74,81,89,20,58,59,88], testosterone [53,60,63,64,70,74,80,81,89,58,59,88], secretory immunoglobulin A (s-IgA) [50,64,66,74,20], alpha-amylase [64,66,77,20], proteins and peptides [52,92,86,51], flow rate [66,20], metabolites [51], osmolality [52,20], creatine kinase and aspartate aminotransferase (AST) [71], potassium and sodium [20], uric acid [65], total antioxidant capacity (TAC) [65], thiobarbituric acid-reactive substances [65], mitochondrial DNA copy number (mtDNAcn) [79].

Cortisol levels decreased in 6 studies [53,63,64,72,74,88], increased in 3 [62,69,70], and remained unchanged in 3 [20,59,81] as a response to induced muscle damage and/or physical fatigue. Other studies quantified cortisol levels as predictors of performance with greater cortisol levels being associated with sporting performance [89] or win probability [58]. Testosterone levels decreased in 3 studies [53,74,58], increased in 4 [63,80,88,59], and remained unchanged in 3 studies [64,70,81]. Like with cortisol levels, several studies investigated testosterone concentration as predictor of performance and/or match outcome finding that both increases in testosterone levels after mid-week training exercises [58,59] as well as lower testosterone on the morning of a match predictive of match success [88]. Several studies examined the testosterone to cortisol ratio (T:C ratio) [61,64,65,71,75,82,20], with 4 reporting an increase [64,65,75,20], 1 reporting a decrease [71], and 2 reporting no change [61,82].

Changes in the expression of proteins and peptides after exercise, physically demanding training, and/or work-related fatigue were explored in 4 studies with one study demonstrating reduced total salivary protein after 5 weeks of military training compared to baseline [52]. Another 2 studies involved the identification of novel fatigue biomarkers from changes in the abundance of peptides/proteins that predicted fatigue status at 83% [51] and 96% [86] accuracy while the last developed a 'fatigue biomarker index' from a ratio of ion intensities from 2 novel peptides that was discriminatory of rating of perceived exertion during an extended exercise trial [92]. In the same study that identified novel protein-based markers of fatigue related to military training, a panel of small molecules (metabolites) were also identified yielding a discriminatory value of 96% [51].

Salivary IgA decreased in 2 studies [50,64] and remained unchanged in 1 [20]. In another, its response varied on the status of the athlete, decreasing if participants finished a 36-hour ultra-endurance event and increasing if they dropped out or were stopped before 24 hours [66]. Alpha-amylase increased in 2 studies [77,20], remained unchanged in 1 [66], and

decreased only at the 8<sup>th</sup> and final mesocycle of a 40-week professional soccer season [64]. Salivary flow rate showed varying responses in both studies investigating its change after exercise, with 1 study showing an increase in females but not males after exercise [20] and the other showing an increasing in non-finishers of a 36-hour endurance event but no change in finishers [66]. Osmolality was increased by approximately 25% at the end of 5 weeks of basic military training [52] but remained unchanged after a single bout of exercise [20]. Rutherfurd-Markwick et al. found that potassium ions (K+) increased for females after exercise but remained unchanged for males, while sodium ions (Na+) did not change for either sex [20]. In response to accentuated eccentric training protocol among resistance trained sports science students, both creatine kinase and aspartate aminotransferase were elevated post-exercise [71]. Uric acid levels remained depressed 38-hours after muscle damage induced by professional soccer games, while total antioxidant capacity remained unchanged and thiobarbituric acid-reactive substances were elevated [65]. Another study examined ventilatory efficiency during an 8-minute cardiopulmonary exercise on a treadmill (cycling 4–6 minutes at 80–90% maximal heart rate) and found that mitochondrial DNA content (mtDNAcn) was inversely correlated with $VE/V_{CO2}$ [79].

**Sleep Deprivation.** Twelve studies measured the response of salivary biomarkers to sleep deprivation [33,49,57,68,78,87,19,61,86,92–94]. These included cortisol [33,57,68,78,87,61,93], testosterone [33,49,57,68,61,93,94], T:C ratio [57,68,61], alpha-amylase [19], DHEA [94], melatonin [87], and untargeted proteins identified via mass spectrometry [92,86]. Cortisol levels were significantly increased with sleep deprivation in 4 studies [57,68,61,93], while in 2 studies cortisol level changed relative to a factor like work-shift time [87] or time of saliva sampling [33]. For example, overnight shift workers showed a trend toward elevated cortisol in the evening compared to day-shift workers [87], and after a 53-hour military training exercise, morning cortisol levers were lower compared to baseline while evening levels were higher [33]. Another study showed a decrease in cortisol awakening response after a 24-hour sleep deprivation protocol [78].

Testosterone levels were decreased in 3 studies [33,49,94] and remained unchanged in 4 [57,68,61,93]. T:C ratio decreased in 2 studies [57,61] and increased in 1 study [68]. Alphas-amylase was found to decrease in 1 study [19] while DHEA was also decreased in another [94]. A single study examined melatonin and reported no change due to sleep deprivation from shift work [87]. One study identified a pair of peptides that could discriminate the sleep deprived group from control [92]. Another proteomic study was able to identify a group of 30 proteins related to fatigue-status and established a model that discriminated between fatigue (after 18–24 hours work shift) and non-fatigue (before work shift) status at 96% accuracy that is likely largely related to shift work-based sleep deprivation [86].

**Dehydration.** Five studies examined dehydration [85,91,75,83,82] with biomarkers investigated including osmolality [85,91,75,83,82], total protein concentration [85,91,75], flow rate [85,75] and alpha -amylase, K+, chloride ions (Cl-), and cortisol [91]. Osmolality [85,91,82,83,95] and total protein concentration was found to increase with dehydration in all studies in which it was tested, while flow rate also consistently decreased [85,95] in response to dehydration. The range increase of osmolality was 38% -110% (pre/post) for active dehydration and 9% for passive dehydration. Total protein increased in the range of 133% - 500%. Meléndez-Gallardo et al. reported that Cl<sup>-</sup> level is effective (p<0.05) in identifying dehydration in even mild changes in body mass of 1.5% [91]. The study also found an increase in levels of cortisol, K+, and alpha-amylase due to moderate (2% water loss) dehydration [91].

**Environmental Stress.** Five studies examined environmental stress. Among them, 4 investigated the impact of high altitude [67,90,18,55], while 1 examined the effects of cold temperature [75]. Markers studied included cortisol [67,90,55], pH [18], DHEA-s [55], IgA, osmolality, and flow rate [75]. High altitude was found to elevate cortisol [67,90,55], DHEA-s [55], and pH levels [18] of saliva following exercise. Another study demonstrated a relationship between ambient temperature and salivary IgA response but found no influence of cold temperatures on saliva flow rate of IgA secretion [75].

**Multi-Stressor.** Eight studies examined multiple stressors with 6 studies examining military populations [33,49,52,53,94,51] and 2 studies involving athletes where exercise was coupled with another stress source including

sleep efficiency [76], and sleep and contextual variables (crowd size, play time, number of matches) [89]. Biomarkers evaluated included cortisol (n=4), testosterone (n=3), DHEA-s (n=1), melatonin (n=1), alpha-amylase (n=1), osmolality (n=1), total protein (n=1), and proteomics and/or metabolites (n=2). Cortisol decreased following field-training but was significantly increased during the 3-day recovery period compared to baseline in one study due to stressors that included military training, exercise, and sleep deprivation [53] while it increased in another due to survival training associated [54]. Lieberman et al. reported a decrease in morning levels of cortisol but an increase in evening levels due to intense exercise with load carriage, sleep deprivation, dehydration, and heat environmental stressors [33]. Another study did not provide specific quantitative concentration levels of cortisol but found an association between cortisol levels and performance ratings by coaches and players [90]. Several studies reported that testosterone [33,53] and T:C ratio [53] decreased in response to military training, exercise, sleep deprivation, and heat environment stressors. Taylor et al. discovered that DHEA-s levels increased during both the morning and afternoon, while the DHEA-s to cortisol ratio exhibited an increase in the morning and a decrease in the evening. [54]. The digestive enzyme alpha-amylase was shown to increase shortly before, and then significantly decrease 1 day after a Taekwondo competition relative to baseline [77]. Fogt et al. reported that osmolality increased while total protein concentration decreased in Air Force basic military trainees due to exercise and dehydration stressors [52]. In one study, a proteomic and metabolomic analysis identified clusters of proteins and metabolites capable of distinguishing between various states of combat missions [51]. Additionally, another study identified clusters of metabolites associated with cognitive performance and their ability to differentiate between groups based on their sleep patterns [76]. A single study examined melatonin response to intense exercise with load carriage, sleep deprivation, dehydration, and heat environment stressors and found no change in concentration levels [33].

## Discussion

The systematic review's purpose was to identify salivary biomarkers sensitive to common stressors experienced by TA in the existing literature. In regard to the first aim, multiple studies have identified salivary biomarkers that are responsive to the acute stressors of muscle damage and physical fatigue, sleep deprivation, dehydration, and environmental stress (see S3 Table). Hormonal biomarkers, such as testosterone and cortisol, were generally found to be sensitive to multiple stressors; however, changes in protein expression and metabolite concentration appear to have greater utility to diagnose response to specific stressors. Our secondary aim was to quantify the impact of acute stressors on the response of salivary biomarkers. However, the interpretation of reported results is challenging due to the heterogeneity of study methodologies, making generalizable claims difficult. In most instances, changes in salivary biomarker levels were associated with reported physical or cognitive performance outcomes. This offers direction for future research aimed at exploring non-invasive approaches to assessing the state of readiness of TA in field settings.

### Salivary Biomarker Response to Acute Stressors

**Muscle Damage and Physiological Fatigue.** The majority of studies examined the response of salivary biomarkers to muscle damage and physiological fatigue, identifying creatine kinase, aspartate aminotransferase, uric acid, cortisol, testosterone, and the T:C ratio as responsive markers. However, the direction of change (e.g., increase or decrease) and magnitude of response of these markers varied widely across studies. For instance, cortisol and testosterone, two of the most frequently measured biomarkers of muscle fatigue and damage, demonstrated mixed responses. Thus, the evidence suggests that hormonal markers like cortisol and testosterone alone may lack the necessary sensitivity and specificity to discriminate TA readiness to perform occupational duties due to their responsiveness to other factors like psychological and social stress [96] and diurnal rhythm [97].

Despite their individual limitations, there was evidence to support their combined utility. One study found that midweek levels of cortisol and testosterone predicted rugby match success with 72.2% accuracy [58]. Another study found a

significant increase (p <0.001) in pre- to post- mid-week T:C ratio before winning games compared to no change before losing games [59], suggesting that the T:C ratio may serve as a more reliable indicator of anabolic/catabolic balance which is reflective of body strain (low ratio) or recovery (high ratio) [98]. Similarly, the ratio of DHEA-s, an anabolic HPA regulator, to cortisol was explored in response to acute hypobaric hypoxia [55]. Despite the rapid increase in cortisol in response to hypoxia, the overall DHEA-s to cortisol ratio remained the same [55], highlighting the potential utility of ratios over single biomarkers for assessing stress resilience.

Other biomarkers frequently studied included IgA [64,66,74,20] and alpha-amylase [64,66,77,91,20], which also exhibited inconsistent directional results across studies. For example, Cook et al. reported decreased IgA levels in ultramarathon *finishers,* while *non-finishers* participating for more than 12 hours showed increased levels [56]. Additional salivary biomarkers like creatine kinase [71], thiobarbituric acid-reactive substances (TBARS) [65] and uric acid [71,65] demonstrated potential utility, albeit with varying response timelines. TBARS, a byproduct of oxidative stress, showed significant increases post-professional soccer games before returning to baseline 48-hours after [65]. In comparison, creatine kinase and uric acid exhibited delayed responses, with significant changes noted 48 [65] and 96 hours [71] post-exercise, respectively. Uric acid in particular is known as a key salivary antioxidant and has shown to reliably indicate stress during physical exertion as prior evidence has demonstrated that resistance training significantly elevates salivary uric acid levels that were well correlated with serum markers of oxidate stress [99].

Innovative approaches involving proteomics and metabolomics have identified promising markers for fatigue detection. For instance, a panel of five proteins (e.g., periplakin, heat shock protein alpha family B member 9, myeloperoxidase, heat shock protein family A member 9, and transketolase) discriminated between fatigued and non-fatigued individuals with 83% accuracy following military training, while a panel of 5 metabolites (e.g., $C_{16}H_{13}$-ClN$_2$O, gamma-butyrobetaine, L-proline, proline-glycine, and pipecolic acid) achieved 96% accuracy in discriminating fatigue status in the same sample [51]. The identified molecules associated with innate immunity, protein cycling and processing, and metabolism of sugars and amino acids, were identified using a discovery approach in which samples were analyzed using mass spectrometry to quantify all detectable proteins and/or metabolites. From which a subset that best predicted fatigue status was identified using statistical techniques like linear discriminant analysis (LDA) [51] and Fisher discriminant analysis [86]. Other recent evidence supports the use of the proteome to evaluate physical stress as proteins linked to immune response, antioxidant activity, and structural integrity were found to be responsive to both exercise and sex [100]. Although not yet practical for field settings, use of multi-omic analyses and machine learning techniques may allow for the discovery of novel combinations of biomarkers that are more robust to some of the common confounders to predictions based off singular analytes like diurnal rhythm or race (e.g., genetics), which can then be used to develop a novel assay.

The greater responsiveness to fatigue of the markers found in these studies may be due to a variety of reasons including their greater antigenicity [86], reduced susceptibility to non-exercise factors such as dietary and health conditions, and perhaps most importantly, the availability of numerous proteins or metabolites involved in a wide variety of physiological pathways. However, these discovered proteomic markers have not been studied as extensively as testosterone and cortisol. Further research is necessary to characterize their responses to physical stress and establish their reliability for operational readiness assessments.

**Sleep Deprivation.** Cortisol levels consistently increased in response to sleep deprivation [57], with a noteworthy decrease in the magnitude of the awakening response (e.g., the rapid cortisol rise 30 minutes after awakening) [78]. These findings align with previous evidence suggesting that sleep deprivation can impair HPA function. This impairment results in elevated evening cortisol levels due to disruptions in glucocorticoid feedback mechanisms [101] and reduced morning, cortisol secretion, reflecting alterations in normal circadian rhythm [102].

In comparison to cortisol, the impact of sleep deprivation on testosterone levels was more variable. While testosterone tended to decrease following sleep deprivation [33,49,94], several studies reported no significant change [56,61]. This inconsistency may be due to methodological differences in factors like population studied, chronic vs acute

sleep-deprivation, lack of standardization in sleep loss protocol, and/or different saliva collection time points. Evidence suggests that the degree of sleep-deprivation is a critical factor where total sleep deprivation appears to reduce serum testosterone levels, while partial sleep-deprivation does not [103]. These trends were mirrored in the reviewed studies examining salivary testosterone as those with more intense sleep deprivation (e.g., from military training) [17,33,49] resulted in decrease testosterone while those with less intense sleep loss did not [56,61]. The precursor to testosterone and other steroid hormones, DHEA-s, also decreased following sleep deprivation from military training [17]. Additionally, decreased levels of alpha-amylase were found in one study after extensive sleep-deprivation and was suggested to be sensitive to the diurnal rhythm for arousal and capable of monitoring performance by closely tracking the diurnal drive for alertness [19].

Proteomics-based studies further highlight the potential of advanced biomarker panels in assessing fatigue related to sleep deprivation. Two studies investigating the impact of sleep deprivation on salivary biomarkers used proteomic markers to predict fatigue status with one identifying clusters of peptides that successfully distinguished between fatigued and non-fatigued groups with over 90% accuracy [92] and another identifying a group of 30 proteins and established a model with 96% accuracy rate in discriminating between subjects fatigued from long-duration shift-work compared to non-fatigued subjects [86]. These findings highlight the potential for proteomics to enhance the precision of fatigue detection caused by sleep deprivation. Panels of proteomic biomarkers may provide a more comprehensive and reliable method to assess sleep quality and/or response to poor sleep, particularly in operational and occupational settings.

**Dehydration.** Research has consistently demonstrated dehydration's detrimental effects on various fitness components, including aerobic performance [104–106], strength [107,108], power [109,110], and balance [111,112]. Additionally, reviews have found that a >2% BW loss impairs cognitive functions related to attention, executive function, and motor coordination [113–115]. This body of evidence supports the need for hydration monitoring to maintain operational readiness in the environments where TAs operate. As TAs frequently operate in environments where both physical exertion and heat exposure are high, early detection and management of hydration status are essential for maintaining both performance and operational effectiveness.

Identification of hydration status using salivary biomarkers showed promised based on the results of the studies included. Salivary properties such as increased osmolality and total protein concentration, and decreased flow rate consistently indicated dehydration, particularly at moderate to severe levels [52,20,83,95]. These relationships are well-supported by physiological mechanisms, as dehydration alters salivary gland function through changes in serum osmolality and volume [116,117]. Additionally, dehydration triggers the release of acute phase proteins and an elevation in globulin concentration [118]. Other biomarkers, such as cortisol and Cl-, also responded to dehydration [91]. Notably, Cl- was particularly effective in detecting mild dehydration (~1.5% decrease of body weight), likely from to its stronger connection to water movement due to its ionic nature This sensitivity suggests Cl- could enable early intervention before more severe dehydration-related effects occur [91]. These findings are especially relevant as hydration is a critical concern for TAs [119], with physical performance declining at dehydration levels of >2% body weight, particularly under heat stress [120].

**Environmental Stress.** The primary environmental stressors addressed in the studies included were temperature and altitude. Altitude-induced stress was associated with elevated levels of cortisol [67,90,55], DHEA-s [55], and salivary pH [18]. Cortisol and DHEA-s levels increase with elevated altitude as a physiological adaptation to the reduced oxygen availability at higher altitudes, prompting adaptation of the HPA axis [121,122]. Salivary pH may also rise due to changes in buffering capacity influenced by hypoxia [123]. While altitude exposure may be less relevant for monitoring the readiness of TAs compared to more common acute stressors like sleep deprivation, dehydration, and muscle fatigue/damage, the physiological effects of high altitude are well documented [124] and can still pose challenges to operational readiness in specific contexts. In addition, the effect of cold temperature was investigated in a single study whereby exercise in cold temperatures relative to normal temperatures resulted in decreased osmolality but not salivary IgA [75]. Although limited, this finding suggests that cold environments may influence certain salivary biomarkers differently than other stressors, warranting further exploration to understand the implications for TA readiness in extreme conditions.

 

**Multi-Stressor.** Several studies incorporated multiple stressors, predominantly within military populations [33,52–54,51], enhancing ecological validity by more accurately reflecting the multifaceted nature of military duties better than studies with single stressors. These findings offer a more comprehensive understanding of the physiological responses to complex, real-world stressors. Notably, inconsistent cortisol responses observed in these studies [33,53,54], highlight the variability of stress reactions in different individuals and situations. The decreases in testosterone levels and the T:C ratio following military training suggest a potential overtraining effects [125–127], which is critical for developing effective training and recovery protocols. Conversely, the increase in DHEA-s levels increased during stressful captivity training, along with the variations in DHEA-s to cortisol ratio between morning and evening, indicates an adaptive response to prolonged stress [54].

Signs of dehydration during military training, evidenced by increased osmolality and decreased protein concentration [52], emphasize the need for proper hydration strategies to maintain TA operational readiness. Lastly, while most studies focused on standard hormonal panels and salivary properties, one study's use of proteomic and metabolomic profiling to identify proteins linked to chronic stress and distinguish acute battle-related stress events underscores the potential for advanced biomarker analysis in understanding stress responses [51].

The collective findings from the studies reviewed have identified several salivary biomarkers, as depicted in Fig 3, which show promise for detecting exposure to specific acute stressors frequently encountered by TA. These salivary bio-markers appear to mirror either a systemic reaction to acute stressors or responses specific to certain conditions. Cortisol, a key marker of HPA axis activation, indicates physiological stress across various types of stressors, and affects nearly every tissue due to the widespread presence of cortisol receptors [10]. Cortisol also influences the levels of other significant biomolecules, such as alpha amylase [128], demonstrating its broad physiological impact. However, while cortisol serves as a general indicator of stress, it may lack the specificity needed to differentiate between individual responses or distinct stressors. Additionally, testosterone levels and the T:C ratio offer additional insights to the anabolic-to-catabolic balance, which is crucial for muscle recovery and adaptation [129]. While cortisol and testosterone reflect systemic responses and deliver valuable information, proteomic analyses provide further insight into cellular processes, revealing more distinct biological markers [130]. This finer resolution aids in understanding the unique signatures associated with different physiological stressors and may guide the development of targeted strategies to monitor and mitigate the effects of acute and chronic stressors in TAs.

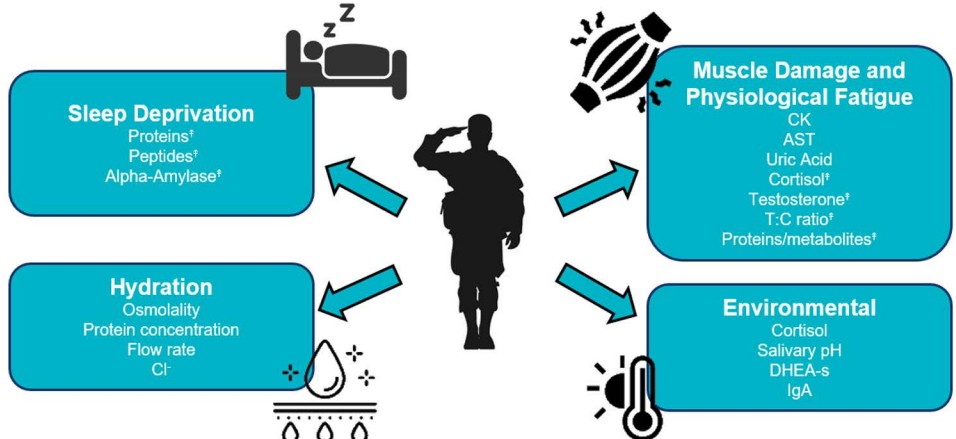

**Fig 3. Commonly reported stressors of Tactical Athletes (TA) and responding salivary biomarkers identified in this review.** ‡Indicates that the salivary biomarker was included in predictive models of outcomes relevant to TA success.

## Salivary Biomarkers of Cognitive and Physical Performance

While the primary objective of this review was to examine salivary biomarker responses to acute stressors, translating these findings into practice requires establishing their associations with metrics of operational readiness. Operational readiness for TAs inherently depends on both cognitive and physical performance. Identifying biomarkers that not only respond to stressors but also predict or correlate with changes in these performance metrics is essential for practical application in monitoring and optimizing TA performance. Thus, one of the secondary aims of the review was to identify if biomarkers were responsive or predictive of physical or cognitive performance measures.

**Cognitive Performance.** Six of the identified studies reported cognitive outcome measures [19,76,78,87,92,131]. Notably, Michael et al. [92] demonstrated that a panel of specific salivary peptides could differentiate between sleep-deprived individuals and controls with an area under the curve (AUC) of 92%, while cognitive performance on tasks such as the Stroop color test, could discriminate between sleep deprived individuals and controls with an AUC of 88%. While these peptides hold potential as a proxy measure of cognitive performance before and after sleep-deprivation, a direct relationship between these biomarkers and cognitive performance was not established [92].

Further insights came from Machi et al. [87], who observed a 28% reduction in the morning cortisol peak following a night shift as well as declines in cognitive performance on the Stroop color word test and the Southern California Repeatable Episodic Memory. Similarly, cortisol levels post-sleep deprivation was related to scores on the Acute Readiness Monitoring Scale (ARMS), particularly its cognitive subscales, which were associated with the psychomotor vigilance test results [78]. In another study, Lieberman et al. [131] found that elevated cortisol levels prior to field military training were inversely related to cognitive reaction times, indicating that higher cortisol may be indicative of cognitive strain under stressful conditions.

Other biomarkers also showed promise in predicting cognitive outcomes. Pajcin et al. [19] identified a positive association between alpha-amylase levels and performance on both the psychomotor vigilance test and driving simulation tasks following sleep deprivation, suggesting alpha-amylase is responsive to cognitive and motor skills under stress. Lastly, Akazawa et al. [76] discovered that several metabolites, including citrulline, were elevated in individuals with poor sleep quality. Citrulline showed a strong inverse correlation with cognitive performance during high-intensity exercises, suggesting its potential as a biomarker for cognitive decline in physically demanding situations. Collectively, the findings suggest that specific salivary biomarkers, including cortisol, alpha-amylase, and citrulline, are responsive to changes in cognitive performance, particularly under conditions of sleep deprivation and high-intensity activity. However, further research is needed to establish direct and consistent relationships between these biomarkers and cognitive performance metrics.

**Physical Performance.** There were more studies that examined physical performance outcomes than cognitive (S3 Table). Testosterone levels increased significantly (91%) in response to a maximal aerobic velocity test [68], likely due to the short-term anabolic effects of high-intensity activity [132]. Similarly, testosterone increased by 20% in men and 15% in women following sprint cycling [80]. Several studies found the T:C ratio declined in response to a maximal aerobic test [61] and after two weeks at a sports climbing camp [70] but was primarily driven by increases in cortisol. Conversely, Mclean et al. [60] found that while post-rugby match countermovement jump performance declined, cortisol and testosterone remained unchanged, suggesting an adaptation to stress in highly trained individuals [133,134]. However, this finding contrasts with results from a 21-day cycling training regimen where daily T:C ratio fluctuated dramatically (0–400%), yet performance remained consistent [74]. Moreover, a number of other studies reported changes in cortisol and testosterone in relation to physical performance outcomes with mixed findings [63,89,58,59]. Such variability underscores the complex and context-dependent relationship between hormonal biomarkers and physical performance.

Non-hormonal biomarkers also showed associations with physical performance in several studies examining the relationship between changes in non-hormonal markers with physical performance [66,79]. Bellar et al. reported that that IgA levels dropped significantly amongst ultramarathon finishers reflecting a decline in mucosal immunity [66]. Additionally, Chen et al. [79] found exercise ventilatory efficiency was associated with mtDNAcn, and may be indicative of aerobic fitness.

Cumulatively, the findings emphasize the complex relationship between hormonal responses to stress and their relationship to physical performance. While hormonal markers such as testosterone, cortisol, and the T:C ratio provide insight into anabolic and catabolic balance, their responsiveness to physical performance outcomes remains inconsistent. Non-hormonal biomarkers like IgA and mtDNAcn may offer additional sensitivity and specificity for monitoring changes in physical readiness. Therefore, future research should explore non-hormonal biomarkers that show evidence of being able to respond to changes in physical performance more sensitively and specifically.

## Implications and Future Research Directions

The findings regarding salivary biomarkers responsiveness to acute stressors demonstrate the feasibility of developing non-invasive, point-of-care operational monitoring technology based on saliva for TA populations. Specifically, the review highlights the potential of proteins and other non-hormonal biomarkers as more promising sensitive and specific signals of stress compared to commonly measured hormonal biomarkers such as cortisol and testosterone. Because cortisol and testosterone are responsive to numerous stressors [135,136] and exhibit diurnal variations [137,138], this presents challenges for their use in precise readiness assessment. Effective diagnostic tools for assessing operational readiness must include high levels of sensitivity and specificity [139]. Consequently, it appears that panels of non-hormonal biomarkers hold the greatest promise as an initial step toward developing technology capable of accurately "diagnosing" the state of operational readiness.

Ultimately, saliva-based technologies may complement other monitoring technologies, such as wearable devices that track physiological parameters (e.g., heart rate, core temperature) [12]. Integrating real-time, continuous measures of biological signals with saliva biomarker assessments may offer a more comprehensive approach to operational readiness monitoring of TAs. Furthermore, saliva assessments have the potential to be incorporated into routine health assessments to facilitate early detection of chronic diseases. For instance, saliva contains a diverse array of biomarkers beyond stress-responsive biomarkers, including markers of immune function, [50] inflammation [72,140], and metabolic activity [140]. Simultaneously assessing multiple biomarkers enables a comprehensive understanding of the physiological impact of stressors on military personnel, enabling tailored preventive interventions to enhance long-term TA health and performance.

However, further research is essential to facilitate the development of field-based technology that uses saliva to inform the state of operational readiness of TAs. Specifically, two areas warrant attention from future research studies. First, it is imperative to increase current knowledge regarding the time course of changes following the onset of stressors. Many studies used a simplistic pre-and-immediately-post design, overlooking the nuanced dynamics of biomarker concentration levels during stress exposure and subsequent recovery patterns and the eventual return to baseline. Future studies should explore the temporal patterns of biomarker responses to acute stressors, capturing the full spectrum of changes from onset to recovery. Understanding these dynamics will be critical for developing robust models that accurately reflect readiness over time.

Secondly, addressing confounding factors that may influence individual responses to acute stressors is paramount. The expression and secretion of these biomarkers can be influenced by factors such as age [141], sex [141], diet [142], and medication [143] can impact the composition and quantity of saliva biomarkers, complicating the interpretation of results. Moreover, considering that many biomarkers demonstrate intra-individual variability [144,145] future research will need to use methodological approaches to better establish baseline biomarker concentration levels, or rather a range of levels which would be considered normal for most individuals.[7] Biomarkers with low daily variation yet responsive to acute stressors would therefore be preferable for assessing operational readiness [7]. Regardless, establishing baseline levels is necessary for the development of predictive models of readiness. Such models must account for intra-individual variability to accurately assess readiness levels.

In summary, individualized models, tailored to the unique biomarker profiles of each TA, are likely necessary for achieving precision in operational monitoring. By advancing the current understanding of salivary biomarkers and addressing

 

these research priorities, saliva-based monitoring systems hold immense potential to evolve into reliable tools for real-time, individualized assessments of operational readiness. These tools, when integrated with other technologies, have the potential to enhance the management of TA health and performance, enabling proactive and targeted interventions to sustain operational capability in demanding environments.

### Correlations between Saliva and Serum Biomarkers

Understanding the correlation of salivary biomarkers levels with those of serum is critical for validating saliva as a reliable medium for monitoring physiological responses. Serum biomarkers are widely considered the gold standard for assessing systemic physiological changes due to their high concentration and direct representation of circulating factors. In military settings, serum markers such as hepcidin [146], brain-derived neurotrophic factor (BDNF) [2,147], and cortisol [2,147] have been strongly linked to performance and resilience outcomes. However, the practicality of serum sampling in operational environments is limited by its invasive nature and logistical challenges.

In contrast, saliva biomarkers offer a less invasive, field-friendly alternative that is particularly useful in operational settings where frequent and non-disruptive sampling is essential. Additionally, since saliva contains less proteins than serum, the risk for non-specific interference is reduced [148]. The extent to which salivary biomarkers correlate with serum biomarkers depends on the molecular properties of the biomarker and the mechanisms governing its transport into saliva. For example, hormones like cortisol and testosterone show strong correlations between serum and saliva levels because they are lipophilic and can passively diffuse through the salivary glands. This makes saliva a reliable medium for tracking stress and recovery in operational scenarios via hormonal markers such as these [149,150].

Although not the focus of the present review, several included studies measured hormonal biomarkers like cortisol and testosterone in both serum and saliva, reporting similar responses to stress across both media. These findings support the utility of salivary hormones as proxies for their serum counterparts, enhancing the feasibility of saliva-based monitoring in field settings. In contrast, non-hormonal biomarkers generally exhibit weaker correlations between serum and saliva concentrations. For example, correlations between serum and saliva were only between -0.034 to 0.212 for concentrations of creatine kinase, a marker of muscle damage. Similar patterns have been observed in studies outside this review for markers such as alpha amylase, where correlations between serum and saliva were weak [151,152]. These differences may be attributed to the complex transport mechanisms and lower concentrations of non-hormonal markers in saliva compared to serum.

Additionally, correlations between serum and saliva biomarkers are influenced by how salivary biomarker concentrations are expressed. A key debate centers on whether salivary biomarker concentrations should be adjusted for flow rate, which is affected by hydration status and sympathetic activity which commonly fluctuates during exercise [153]. This adjustment warrants consideration because higher flow rates can dilute biomarker concentrations while lower flow rates can result in apparent increased concentrations. Research findings on this issue have been mixed. In one study of creatine kinase after a futsal match, the enzyme remained elevated in both serum and saliva for 36 hours when measured directly. However, when adjusted for flow rate, salivary levels showed no significant change from pre-match levels, though the adjusted values did correlate better with serum levels 12 hours post-match [153]. Another study found that uric acid showed stronger serum-saliva correlations without any adjustment for flow rate or total protein [99]. In contrast, for salivary IgA , flow rate adjustment is widely accepted as necessary to account for variations in saliva volume [154]. These findings suggest that the optimal method for expressing salivary biomarker levels varies by marker and that there may be no universal approach that maximizes serum-saliva correlations across all.

### Limitations

There are several limitations of the available literature. Although a majority of the included articles (55%) were scored as 'good' quality via the Downs and Black checklist, a large portion (45%) were only rated as 'fair' quality with most articles

scoring poorly on external validity and control of confounding factors. This reflects the largely quasi-experimental nature of the included, with most using non-random sampling plans and group assignments.

Heterogeneity in study populations, methodological designs, and biomarkers examined further limits the generalizability of findings. For instance, the effects observed on testosterone levels following mid-week rugby training [88] may not be directly comparable to those from a 21-day cycling training protocol [74], despite both being stressful physical activities potentially inducing muscle damage. Broad methodological shortcomings also include an insufficient consideration of baseline conditions, such as failing to account for diurnal variations in hormone levels before assessing stressor impacts. Additionally, many studies only take a single post-stressor measurement, which does not adequately capture the duration of the stressor's effects. Furthermore, analyses often aggregate data at the group level, which may obscure the wide range of individual differences in biochemical responses.

A more specific methodological limitation was variations in assay designs may significantly impact the measurement and interpretation of biomarker availability across studies. Differences in detection methods, such as enzyme-linked immunosorbent assays (ELISA) versus radioimmunoassays (RIA), influence sensitivity and specificity. For example, ELISA, commonly used in many studies, are scalable and suitable for a broad range of concentrations but can exhibit variable sensitivity depending on assay design [155]. RIA offers superior sensitivity for measuring hormones that are present in low concentrations in saliva, such as cortisol and testosterone [156], but is less accessible due to its reliance on radioactive reagents [156]. These methodological differences can result in discrepancies in biomarker availability, especially for analytes with subtle physiological fluctuations. Most studies included in this review used commercial ELISA kits, introducing variability in sensitivity, specificity, and cross-reactivity, which complicates direct comparisons and may affect the generalizability of findings. Advanced methods, such as liquid chromatography-mass spectrometry (LC-MS) and bicinchoninic acid (BCA) protein assays, real-time polymerase chain reaction (RT-PCR), colorimetric enzymatic assays and kinetic assays were used to quantify protein and/or peptide markers, mitochondrial DNA copy number, and other biomarkers like alpha-amylase and secretory IgA. These techniques are recognized for their high sensitivity and specificity, particularly for complex and/or low-abundance biomarkers such as proteins, enzymes, and metabolites.

Lastly, a significant gap in the research is the underrepresentation of females. Only 13 studies included both male and female participants [53,66,67,69,71,76,78,79,87,90,19,86,20], with a single study involving a military population that included females [53]. This is notable, especially given the physiological and biochemical differences between sexes that could influence biomarker responses. For example, one study specifically designed to explore sex differences found significant increases in osmolality, alpha-amylase activity, and secretory IgA rates in females during steady state exercise, whereas males showed no significant changes [20]. Similarly, following a rugby seven match, a greater increase was observed in muscle injury markers (CK, AST, and LDH) among men compared to women [148] while another study performing proteomic analysis of saliva demonstrated a difference in the proteome response between men and women following a bout of resistance exercise leading to failure [100]. Addressing this gap is particularly important in military contexts, where all combat roles have been open to females since 2016 [157].

## Conclusion

In conclusion, this review highlights the responsiveness of various salivary biomarkers to acute stressors experienced by TAs. While common hormonal biomarkers like cortisol and testosterone provide some insights, their variability and generalized responses highlight the need for a broader panel that include non-hormonal markers to improve the accuracy and specificity of readiness assessments. The review also demonstrates the potential of proteomic approaches to achieve a finer resolution in identifying fatigue and readiness, suggesting that panels of proteomic biomarkers could more effectively gauge the physiological state of TAs in response to diverse stressors.

Future research should prioritize validating salivary biomarkers in larger, more diverse populations and refining the technology for practical field applications. Well-controlled studies are essential to fully realize the potential of salivary

biomarker assessments for operational readiness, with a focus on enhancing our understanding of non-hormonal biomarkers, exploring the temporal dynamics of their responses to stressors, and addressing confounding factors that influence individual responses. The knowledge gained from these studies will be crucial for developing accurate models that tailor biomarker responses to specific stressors and individual variability, thereby advancing the practical applications of salivary biomarker assessments in TA operational readiness monitoring.

## Supporting information

**S1 Table. Search strategy including list of search terms and filters.**
(DOCX)

**S2 Table. Table of all articles identified during literature search process with inclusion/exclusion decision and justification.**
(DOCX)

**S3 Table. Characteristics and key findings of included studies.**
(DOCX)

**S1 Data. Check list.**
(DOCX)

## Acknowledgements

The authors would like to express their gratitude to Dr. Shane Caswell and Dr. Emanuel Petricoin for feedback and support during various stages of the manuscript development.

## Author contributions

**Conceptualization:** Yosef Shaul, Joel Martin.

**Data curation:** Bryndan Lindsey, Yosef Shaul.

**Formal analysis:** Bryndan Lindsey, Yosef Shaul.

**Investigation:** Yosef Shaul.

**Methodology:** Yosef Shaul, Joel Martin.

**Project administration:** Joel Martin.

**Resources:** Joel Martin.

**Supervision:** Bryndan Lindsey, Joel Martin.

**Writing – original draft:** Bryndan Lindsey, Yosef Shaul, Joel Martin.

**Writing – review & editing:** Bryndan Lindsey, Yosef Shaul, Joel Martin.

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
