## [Decision Letter · Decision Letter 0]

6 Dec 2024

PONE-D-24-34053Salivary biomarkers of tactical athlete readiness: A systematic reviewPLOS ONE

Dear Dr.  Martin,

Thank you for submitting your manuscript to PLOS ONE. After careful consideration, we feel that it has merit but does not fully meet PLOS ONE’s publication criteria as it currently stands. Therefore, we invite you to submit a revised version of the manuscript that addresses the points raised during the review process.

We look forward to receiving your revised manuscript.

Kind regards,

Juan Luis Castillo-Navarrete, Ph.D.

Academic Editor

PLOS ONE

Journal Requirements:

4. Please include a copy of Table 3 which you refer to in your text on page 24.

5. As required by our policy on Data Availability, please ensure your manuscript or supplementary information includes the following: A numbered table of all studies identified in the literature search, including those that were excluded from the analyses. For every excluded study, the table should list the reason(s) for exclusion. If any of the included studies are unpublished, include a link (URL) to the primary source or detailed information about how the content can be accessed. A table of all data extracted from the primary research sources for the systematic review and/or meta-analysis. The table must include the following information for each study: Name of data extractors and date of data extraction Confirmation that the study was eligible to be included in the review. All data extracted from each study for the reported systematic review and/or meta-analysis that would be needed to replicate your analyses. If data or supporting information were obtained from another source (e.g. correspondence with the author of the original research article), please provide the source of data and dates on which the data/information were obtained by your research group. If applicable for your analysis, a table showing the completed risk of bias and quality/certainty assessments for each study or outcome. Please ensure this is provided for each domain or parameter assessed. For example, if you used the Cochrane risk-of-bias tool for randomized trials, provide answers to each of the signalling questions for each study. If you used GRADE to assess certainty of evidence, provide judgements about each of the quality of evidence factor. This should be provided for each outcome. An explanation of how missing data were handled. This information can be included in the main text, supplementary information, or relevant data repository. Please note that providing these underlying data is a requirement for publication in this journal, and if these data are not provided your manuscript might be rejected.

Additional Editor Comments:

The manuscript provides a thorough and relevant review of salivary biomarkers for assessing the readiness of tactical athletes. However, several areas would benefit from refinement to improve clarity and strengthen its impact.

First, the **discussion section**  should be restructured into subsections based on stressor types (e.g., dehydration, sleep deprivation, muscle damage), making the narrative more accessible. Incorporating the references suggested by the reviewers, particularly on oxytocin, muscle enzymes, uric acid, and proteomics, would deepen the analysis and enhance the context of the findings.

Second, the **methodology**  requires more detail about the assays used, including sensitivity, specificity, and limitations. Addressing how assay design might influence biomarker variability would improve transparency and reliability.

Third, consider a brief mention of serum biomarkers as a complementary approach, including any known correlations with salivary markers, to provide a broader perspective.

Finally, variability in responses for key biomarkers like cortisol and testosterone should be discussed further, including external factors such as diurnal rhythms or confounding variables like race. Additionally, the proteomics findings merit more detail, particularly regarding experimental methods and data analysis.

These revisions would enhance the manuscript’s clarity, depth, and applicability, and I look forward to reviewing the improved version of this promising work.

Reviewers' comments:

Reviewer's Responses to Questions

**Comments to the Author**

1. Is the manuscript technically sound, and do the data support the conclusions?

Reviewer #1: Yes

Reviewer #2: Yes

Reviewer #3: Partly

2. Has the statistical analysis been performed appropriately and rigorously? 

Reviewer #1: N/A

Reviewer #2: Yes

Reviewer #3: Yes

3. Have the authors made all data underlying the findings in their manuscript fully available?

Reviewer #1: Yes

Reviewer #2: Yes

Reviewer #3: Yes

4. Is the manuscript presented in an intelligible fashion and written in standard English?

Reviewer #1: Yes

Reviewer #2: Yes

Reviewer #3: Yes

5. Review Comments to the Author

Reviewer #1: The manuscript exhibits a solid structure, with a clear description of the methodological design. The results are presented in a manner that aligns seamlessly with the proposed objectives. The discussion addresses the significant challenges in the search for predictive biomarkers through the collection of data based on the analysis of highly diverse experimental designs. However, the authors provide a critical and well-developed contribution to the available knowledge. I agree with the authors on the great potential of proteomic studies for the identification of salivary biomarkers in future experimental approaches focused on the readiness of tactical athletes.

Reviewer #2: This manuscript shows a detailed literature review of use of salivary biomarkers to assess common stressors faced by tactical athletes in various environmental conditions. The use of biomarkers for asessing health of tactical atheletes and millitary personnel is very much needed and this review addresses this need in a highly descriptive and comprehensive way. There are a few concerns that I have listed below.

1) The authors mention that salivary biomarkers have an advantage over serum biomarkers since it is relatively non-invasive. However, serum biomarkers cannot be completely excluded out. A short paragraph on use of serum biomarkers to assess impact of stressors on operational readiness might be highly useful. Also, any correlation with saliva-based studies can be provided wherever information is available.

2) Little information has been provided on the assays used for measuring biomarker levels. Assay design plays a crucial role in biomarker performance and hence information on such should be provided wherever applicable.

3) The authors mention that levels of cortisone and testerone varied considerably across studies probably due to variations in psycological and social stress. One more variability can also be the assay design and as alluded to earlier should be specifically mentioned as a factor that will infuence the outcomes of these tests.

4) More information on the proteomic studies can be given wherever applicalble. For example what kind of proteomics experiment was done and instruments and methods of ionization used and data analysis etc.

5) The impact of race on these kinds of these studies is crucial as it may be a confounding variable. Any information on impact of race on various biomarker test outcomes would be useful

Reviewer #3: This manuscript is interesting, somethings that could improve it would be:

*Try to be more clear and direct in the discussion section

*Evaluate to include addition references that could give interesting data about the role of different biomarkers such as:

Oxytocin:

https://pubmed.ncbi.nlm.nih.gov/38635553/

Muscle enzymes:

https://pubmed.ncbi.nlm.nih.gov/33167318/

https://pubmed.ncbi.nlm.nih.gov/28480688/

Uric acid:

https://pubmed.ncbi.nlm.nih.gov/31514287/

Proteome in general:

https://pubmed.ncbi.nlm.nih.gov/31881350/

6. PLOS authors have the option to publish the peer review history of their article (what does this mean? ). If published, this will include your full peer review and any attached files.

**Do you want your identity to be public for this peer review?** For information about this choice, including consent withdrawal, please see our Privacy Policy .

Reviewer #1: No

Reviewer #2: No

Reviewer #3: No

---

## [Author Response · Author response to Decision Letter 1]

13 Jan 2025

Dr. Castillo-Navarrete and reviewers,

We are pleased to re-submit the following manuscript titled: “Salivary biomarkers of tactical athlete readiness: A systematic review”. With our resubmission we are providing a revised manuscript with changes noted in track changes, a clean verision of the revised manuscript and this response letter. Point by point responses to comments are provided below. We would like to express our appreciation to each of the reviewers and the editor for their time.

Journal Requirments

Response: We have reviewed the manuscript to ensure it meets PLOS ONE’s style requirements.

Response: We have included the title page as part of the main document with the revision and apologize for our misunderstanding previously.

Response: A data availability statement has been added in the submission form with our revision.

4. Please include a copy of Table 3 which you refer to in your text on page 24.

Response: This was a typo, it was changed to table 2.

5. As required by our policy on Data Availability, please ensure your manuscript or supplementary information includes the following: A numbered table of all studies identified in the literature search, including those that were excluded from the analyses. For every excluded study, the table should list the reason(s) for exclusion. If any of the included studies are unpublished, include a link (URL) to the primary source or detailed information about how the content can be accessed.

Response: We have added this as a supplementary table.

A table of all data extracted from the primary research sources for the systematic review and/or meta-analysis. The table must include the following information for each study: Name of data extractors and date of data extraction Confirmation that the study was eligible to be included in the review. All data extracted from each study for the reported systematic review and/or meta-analysis that would be needed to replicate your analyses.

Response: Thank you for the feedback. While we did not conduct a meta-analysis, we included a comprehensive table with the characteristics of the included studies, which contains all the extracted information used in our synthesis. Due to its length, we opted to present this table as a supplementary file rather than in the main manuscript. However, if you feel that including it as a main table would enhance the presentation of the manuscript, we would be happy to make that adjustment.

If data or supporting information were obtained from another source (e.g. correspondence with the author of the original research article), please provide the source of data and dates on which the data/information were obtained by your research group. If applicable for your analysis, a table showing the completed risk of bias and quality/certainty assessments for each study or outcome. Please ensure this is provided for each domain or parameter assessed. For example, if you used the Cochrane risk-of-bias tool for randomized trials, provide answers to each of the signalling questions for each study. If you used GRADE to assess certainty of evidence, provide judgements about each of the quality of evidence factor. This should be provided for each outcome. An explanation of how missing data were handled. This information can be included in the main text, supplementary information, or relevant data repository. Please note that providing these underlying data is a requirement for publication in this journal, and if these data are not provided your manuscript might be rejected.

Response: Thank you for the comment. In our manuscript, we have included a bias and quality assessment table (Table 1), which utilizes a modified Downs and Black checklist. This table presents the total scores as well as scores for each parameter assessed. If further details or additional information are required, we would be happy to provide them as supplementary material or in a data repository to meet the journal's requirements.

Additional Editor Comments:

The manuscript provides a thorough and relevant review of salivary biomarkers for assessing the readiness of tactical athletes. However, several areas would benefit from refinement to improve clarity and strengthen its impact.

First, the discussion section should be restructured into subsections based on stressor types (e.g., dehydration, sleep deprivation, muscle damage), making the narrative more accessible. Incorporating the references suggested by the reviewers, particularly on oxytocin, muscle enzymes, uric acid, and proteomics, would deepen the analysis and enhance the context of the findings.

Response: We appreciate the editor’s suggestion to restructure the discussion section into subsections based on stressor types, as this enhances the clarity and accessibility of the narrative. In response, we have reorganized the discussion into distinct subsections, including Dehydration, Sleep Deprivation, and Muscle Damage and Physiological Fatigue, with each subsection addressing the relevant findings and interpretations for the respective stressor. Additionally, grammatical modifications were made throughout the discussion to improve readability and ensure a more cohesive presentation of the results. These changes aim to provide a more structured and accessible discussion for readers.

Second, the methodology requires more detail about the assays used, including sensitivity, specificity, and limitations. Addressing how assay design might influence biomarker variability would improve transparency and reliability.

Response: We added a column in table 2, ‘Salivary Biomarker Quantification Method’, that lists assay information like sensitivity, CV, minimum limit of detection where available in each article as well as quantification methods of other techniques for biomarkers like proteins, peptides etc. We also added content in the limitations section (line 767-786) addressing how differences in assay design may affect biomarker variability, and therefore, reliability and generalizability of findings.

Third, consider a brief mention of serum biomarkers as a complementary approach, including any known correlations with salivary markers, to provide a broader perspective.

Response: We have added a section (line 718-738) on the correlation of salivary markers to serum markers to discuss how closely salivary biomarkers are known to match to expression in serum.

Finally, variability in responses for key biomarkers like cortisol and testosterone should be discussed further, including external factors such as diurnal rhythms or confounding variables like race. Additionally, the proteomics findings merit more detail, particularly regarding experimental methods and data analysis.

Response: We have added content (line 515-523) regarding the identification of these proteins/metabolites vis multi-omics and discovery proteomics. We agree that these techniques deserve more attention in this field to perhaps find more robust markers of readiness compared to what has been predominantly employed/studied.

Reviewer #1

The manuscript exhibits a solid structure, with a clear description of the methodological design. The results are presented in a manner that aligns seamlessly with the proposed objectives. The discussion addresses the significant challenges in the search for predictive biomarkers through the collection of data based on the analysis of highly diverse experimental designs. However, the authors provide a critical and well-developed contribution to the available knowledge. I agree with the authors on the great potential of proteomic studies for the identification of salivary biomarkers in future experimental approaches focused on the readiness of tactical athletes.

Response: Thank you for your thoughtful feedback on our manuscript. We appreciate your recognition of the methodological design, alignment of results with our objectives, and the critical perspective provided in our discussion. Your acknowledgment of the potential of proteomic studies for identifying salivary biomarkers in tactical athletes reinforces the importance of this work. We sincerely value the time reviewing our study.

Reviewer #2

This manuscript shows a detailed literature review of use of salivary biomarkers to assess common stressors faced by tactical athletes in various environmental conditions. The use of biomarkers for asessing health of tactical atheletes and millitary personnel is very much needed and this review addresses this need in a highly descriptive and comprehensive way. There are a few concerns that I have listed below.

1) The authors mention that salivary biomarkers have an advantage over serum biomarkers since it is relatively non-invasive. However, serum biomarkers cannot be completely excluded out. A short paragraph on use of serum biomarkers to assess impact of stressors on operational readiness might be highly useful. Also, any correlation with saliva-based studies can be provided wherever information is available.

Response: Thank you for this suggestion, we absolutely agree its important to discuss the utility of serum based biomarkers and their correlation with salivary counterparts. We have added language both in the introduction (line 85 – 99). We have also added content in the discussion on this topic.

2) Little information has been provided on the assays used for measuring biomarker levels. Assay design plays a crucial role in biomarker performance and hence information on such should be provided wherever applicable.

Response: We have added a column in our extracted data table called ‘Salivary Biomarker Quantificaiton Method’ which outlines how each marker was quantified. Additionally we added a section in the results that described the assay types used across the studies and general sensitivity and CV (of what was reported).

3) The authors mention that levels of cortisone and testerone varied considerably across studies probably due to variations in psycological and social stress. One more variability can also be the assay design and as alluded to earlier should be specifically mentioned as a factor that will infuence the outcomes of these tests.

Response: This is an excellent point. We have added to the limitations section language to address this source of variation.

4) More information on the proteomic studies can be given wherever applicalble. For example what kind of proteomics experiment was done and instruments and methods of ionization used and data analysis etc.

Response: We have added more content regarding the type of proteomics experiments conducted on line 542-549.

5) The impact of race on these kinds of these studies is crucial as it may be a confounding variable. Any information on impact of race on various biomarker test outcomes would be useful

Response: We appreciate the reviewer highlighting the potential impact of race on biomarker outcomes. Many studies included in our review did not account for genetic or demographic factors, such as race, in their analyses. We have acknowledged this as a limitation and emphasized the need for future research to address this gap.

Reviewer #3: This manuscript is interesting, somethings that could improve it would be:

*Try to be more clear and direct in the discussion section

Response: Thank you for your feedback and for recognizing the potential value of our manuscript. In response to your suggestion, we have revised the discussion section to enhance clarity and directness. Specifically, we have 1) Restructured the discussion into subsections based on specific stressors (e.g., dehydration, sleep deprivation, muscle damage) to improve readability and logical flow; 2) Simplified and streamlined sentences to ensure the narrative is more concise and accessible to readers; and 3) Enhanced transitions between ideas to create a cohesive and focused discussion.

*Evaluate to include addition references that could give interesting data about the role of different biomarkers such as:

Oxytocin:

https://pubmed.ncbi.nlm.nih.gov/38635553/

Muscle enzymes:

https://pubmed.ncbi.nlm.nih.gov/33167318/

https://pubmed.ncbi.nlm.nih.gov/28480688/

Uric acid:

https://pubmed.ncbi.nlm.nih.gov/31514287/

Proteome in general:

https://pubmed.ncbi.nlm.nih.gov/31881350/

Response: Thank you for the suggestion. We have found it useful to add information from the two articles studying uric acid and the proteome as these two markers were identified in our search studies. We have added context from these mentioned articles on uric acid (line 640) and the proteome (line 702).

---

## [Editor Report · Decision Letter 1]

22 Jan 2025

PONE-D-24-34053R1Salivary biomarkers of tactical athlete readiness: A systematic reviewPLOS ONE

Dear Dr. Martin,

Thank you for submitting your manuscript to PLOS ONE. After careful consideration, we feel that it has merit but does not fully meet PLOS ONE’s publication criteria as it currently stands. Therefore, we invite you to submit a revised version of the manuscript that addresses the points raised during the review process. Please submit your revised manuscript by Mar 08 2025 11:59PM. If you will need more time than this to complete your revisions, please reply to this message or contact the journal office at plosone@plos.org . Please include the following items when submitting your revised manuscript:

We look forward to receiving your revised manuscript.

Kind regards,

Juan Luis Castillo-Navarrete, Ph.D.

Academic Editor

PLOS ONE

Journal Requirements:

**Additional Editor Comments:**

After reviewing the changes and the reviewers’ comments, we appreciate the effort you have put into improving the paper. The study addresses a relevant topic, and the revisions have significantly strengthened the manuscript.

To further refine the work, we recommend minor revisions. These include clarifying and streamlining the discussion, incorporating additional references on biomarkers such as oxytocin, muscle enzymes, uric acid, and proteomics, and briefly addressing the role of serum biomarkers and their potential correlations with salivary studies. Providing more details on assays and proteomic methods, as well as discussing the impact of race on biomarker outcomes, would also enhance the manuscript.

We look forward to receiving your revised version and are confident these adjustments will further improve the quality of the work.

---

## [Author Response · Author response to Decision Letter 2]

21 Feb 2025

We are pleased to re-submit the following manuscript titled: “Salivary biomarkers of tactical athlete readiness: A systematic review”. With our resubmission we are providing a revised manuscript with changes noted in track changes, a clean verision of the revised manuscript and this response letter. Point by point responses to comments are provided below. We would like to express our appreciation to each of the reviewers and the editor for their time.

Response: We have reviewed the reference list and made sure that the references are correct and no retracted articles were cited.

Additional Editor Comments:

After reviewing the changes and the reviewers’ comments, we appreciate the effort you have put into improving the paper. The study addresses a relevant topic, and the revisions have significantly strengthened the manuscript.

Response: Thank you for your positive feedback and recognition of the effort put into the revisions. We appreciate your guidance in strengthening the manuscript throughout the review process.

To further refine the work, we recommend minor revisions. These include clarifying and streamlining the discussion, incorporating additional references on biomarkers such as oxytocin, muscle enzymes, uric acid, and proteomics, and briefly addressing the role of serum biomarkers and their potential correlations with salivary studies. Providing more details on assays and proteomic methods, as well as discussing the impact of race on biomarker outcomes, would also enhance the manuscript.

Response: We appreciate the further feedback on our manuscript. To address these points we made further edits to the discussion to improve the structure and flow from paragraph to paragraph. In some cases edits were made to remove any redundant statements.

A few edits were made to address the role of serum biomarkers section of the discussion. The edits were made to highlight their comparative strengths and limitations alongside salivary biomarkers.

Minor edits regarding the assays and proteomic methods were made to results, discussion and supplementary table 3 to ensure consistent language across these components of the paper.

Lastly the additional references were incorporated to the discussion:

González Fernández, Á., de la Rubia Ortí, J. E., Franco-Martinez, L., Ceron, J. J., Mariscal, G., & Barrios, C. (2020). Changes in salivary levels of creatine kinase, lactate dehydrogenase, and aspartate aminotransferase after playing rugby sevens: the influence of gender. International journal of environmental research and public health, 17(21), 8165.

Barranco, T., Tvarijonaviciute, A., Tecles, F., Carrillo, J. M., Sánchez-Resalt, C., Jimenez-Reyes, P., ... & Cugat, R. (2017). Changes in creatine kinase, lactate dehydrogenase and aspartate aminotransferase in saliva samples after an intense exercise: a pilot study. The Journal of sports medicine and physical fitness, 58(6), 910-916.

González-Hernández, J. M., Franco, L., Colomer-Poveda, D., Martinez-Subiela, S., Cugat, R., Cerón, J. J., ... & Tvarijonaviciute, A. (2019). Influence of sampling conditions, salivary flow, and total protein content in uric acid measurements in saliva. Antioxidants, 8(9), 389.

Bishop NC, Gleeson M. Acute and chronic effects of exercise on markers of mucosal immunity. Frontiers in Bioscience. 2009;14(2):4444–56.

González Fernández, Á., de la Rubia Ortí, J. E., Franco-Martinez, L., Ceron, J. J., Mariscal, G., & Barrios, C. (2020). Changes in salivary levels of creatine kinase, lactate dehydrogenase, and aspartate aminotransferase after playing rugby sevens: the influence of gender. International journal of environmental research and public health, 17(21), 8165.

Franco-Martínez, L., González-Hernández, J. M., Horvatić, A., Guillemin, N., Cerón, J. J., Martínez-Subiela, S., ... & Reyes, P. J. (2020). Differences on salivary proteome at rest and in response to an acute exercise in men and women: a pilot study. Journal of proteomics, 214, 103629.

---

## [Editor Report · Decision Letter 2]

3 Mar 2025

Salivary biomarkers of tactical athlete readiness: A systematic review

PONE-D-24-34053R2

Dear Dr.Joel Martin,

We’re pleased to inform you that your manuscript has been judged scientifically suitable for publication and will be formally accepted for publication once it meets all outstanding technical requirements.

Kind regards,

Juan Luis Castillo-Navarrete, Ph.D.

Academic Editor

PLOS ONE

Additional Editor Comments (optional):

Congratulations on your revisions. Your work is highly relevant and makes a valuable contribution to scientific knowledge. The manuscript now meets PLOS ONE’s guidelines, with all necessary corrections successfully implemented.
---

## [Editor Report · Acceptance letter]

PONE-D-24-34053R2

PLOS ONE

Dear Dr. Martin,

I'm pleased to inform you that your manuscript has been deemed suitable for publication in PLOS ONE. Congratulations! Your manuscript is now being handed over to our production team.

Kind regards,

on behalf of

Dr. Juan Luis Castillo-Navarrete

Academic Editor

PLOS ONE